# Impact of Drying on Structural Performance of Reinforced Concrete Beam with Slab

**DOI:** 10.3390/ma14081887

**Published:** 2021-04-10

**Authors:** Pranjal Satya, Tatsuya Asai, Masaomi Teshigawara, Yo Hibino, Ippei Maruyama

**Affiliations:** Department of Environmental Engineering and Architecture, Graduate School of Environmental Studies, Nagoya University, Nagoya 464 8603, Japan; satya.pranjal@e.mbox.nagoya-u.ac.jp (P.S.); teshi@isc.chubu.ac.jp (M.T.); hibino@nuac.nagoya-u.ac.jp (Y.H.)

**Keywords:** reinforced concrete, beam, slab, stiffness, yielding, moment-curvature relationship, drying, shrinkage, drying cracks

## Abstract

Evaluating the performance of reinforced concrete (RC) structures during earthquakes and the resultant damage in the structures depends on an accurate load–displacement relationship. Several experimental and analytical evaluation methods for load–displacement relationships have been proposed and specified in current design standards. However, there have been few quantitative studies on the impact of drying on the yielding behavior of RC members, including evaluations of the effective stiffness of members. In this study, to investigate changes in the mechanical properties of RC beam–slab members due to drying of the concrete, cyclic loading tests are conducted on two RC beam–slab members with and without drying. It is found that the lateral structural stiffness of the specimen with drying decreased to 77% that of the specimen without drying. This is verified in the calculation of the flexural stiffness. In this calculation, it is assumed that drying shrinkage decreases the moment of inertia of the slab in tension but not in compression. Meanwhile, no difference is observed in the flexural capacity and yield displacement between the two specimens. Thus, there is no significant impact from drying shrinkage in RC beam–slab members on the lateral structural performance, while the shrinkage instead induces greater flexural cracking, which reduces the residual stresses in the specimen with drift leading to a gradual decrease in the impact of drying.

## 1. Introduction

Evaluating the responses of reinforced concrete (RC) structures during earthquakes and the resultant damage in the structures depends on an accurate load–displacement relationship. The load–displacement relationship is controlled by the stiffness of RC members, which relies on several other factors such as the geometry of the section, material properties, and acting load. Much study has focused on understanding and defining the effective stiffness of RC structures [1,2,3,4,5]. To determine an accurate load–displacement relationship, significant efforts have been undertaken, and empirical and theoretical evaluation methods for load–displacement relationships have been proposed in experimental and analytical studies [6,7,8] and have been specified in current design codes [9,10,11,12]. These evaluation methods have focused on seismic behavior without considering any change in material properties after casting.

However, studies by the authors have shown that the frequency of concrete structures decreases (i.e., the stiffness of the structure decreases) with the change in material properties with age and for different curing conditions [13,14,15,16]. To prevent the failure of RC buildings, any change in the mechanical properties of the structures due to changes in the material properties should be investigated and considered in design methods.

For concrete mixtures with an ordinary concrete strength level, the amount of water used is greater than that required for the hydration of ordinary Portland cement in order to achieve better workability. Consequently, even 28 days after casting, the internal relative humidity is quite high, and if the concrete is exposed to a typical environment, it starts to dry and shrink. The hardened cement paste (hcp), which has a binder role in concrete, shrinks under drying. While the mechanism for this phenomenon is still under discussion, capillary tension, surface energy, disjoining pressure [17,18,19,20], and the irreversible alteration of C–S–H [21,22] are thought to play an important role in the shrinkage of hcp.

Aggregates in concrete exhibit relatively less shrinkage than that of hcp under a drying condition. Considering the larger Young’s modulus of aggregates as compared to that of hcp, the aggregates in concrete restrain the shrinkage of hcp. Consequently, cracks occur around the aggregate [23,24]. Shrinkage is affected by aggregate size and gradation [25,26,27,28] as well as by type of aggregates used [29,30]. As the concrete shrinkage is less than that of the hcp, the shrinkage strain restrained by the aggregate is responsible for cracks opening in the concrete [31], and thus, the Young’s modulus of the concrete is decreased to 60% of that in the sealed condition after drying in a typical environment [32]. This reduction in concrete stiffness under drying might contribute to the structural stiffness reduction of RC members [33,34].

For RC structures, the drying shrinkage of concrete is restrained by the reinforcing bars and other connecting RC members. This results in a tensile stress leading to cracking in the concrete, which also reduces the structural stiffness of the RC member. Tanimura et al. [33] proposed methods for estimating the performance of RC beams with autogenous shrinkage under a bending load. The shear capacity of RC beams affected by autogenous shrinkage [35] and drying shrinkage [36] has been studied. Sasano et al. [34] experimentally and analytically investigated the effect of drying shrinkage on the load–displacement relationship of RC shear walls, which are relatively more prone to drying shrinkage than beams and columns. They found that the drying of concrete reduced the initial structural stiffness by approximately 54%, while the yield displacement of the wall was not significantly altered.

Based on these results, similar to the results for RC shear walls, the effect of drying must play a significant role in RC beam–slab members. However, there have been few quantitative studies on the mechanical properties of beam–slab members with concrete drying. The purpose of this study is to investigate changes in the mechanical properties of RC beam–slab members due to the drying of concrete. Cyclic loading tests are conducted on two specimens with varying curing conditions, with and without drying. One specimen is tested immediately after long-term curing in water, and the other specimen is tested after curing with exposure to the air so that the concrete dries sufficiently. The authors [37] have already discussed the initial lateral structural stiffness of an RC beam–slab member without drying and reported that yield displacement can be defined as occurring where high-energy absorption is demonstrated when the main bars of the beam–slab member yield. In this study, furthering the results of the previous study, changes in the seismic behavior of an RC beam–slab member with drying, primarily the structural stiffness and yield deformation, are discussed by comparing the experimental results, as drying is related to surface exposed to the environment per volume. Slabs used in this study with the beams have a larger surface/volume ratio rather than those of beams, which indicates quick and significant amount of drying due to the relative different drying-induced shrinkage strain of the members, restraint drying shrinkage—induced crack may occur in slabs after the drying. This makes it ideal for clearly understanding the impact of drying on structural performance of beams jointed with slabs. Furthermore, due to constraints of laboratory and a lack of interest on the structural drying impacts, experimental studies of reinforced concrete structures after long term drying are limited. Therefore, this study plays an important role in experimentally understanding the impact of drying on actual reinforced concrete structures.

## 2. Experimental Outline

### 2.1. Specimen Properties

Two identical specimens were developed to represent a 1 span × 1 span floor at a one-third scale. The specimen has two large beams sandwiching two slabs and one sub-beam. The large beam has a 200 mm × 300 mm cross section and that of the sub-beam is 150 mm × 250 mm. The thickness of the slab is 80 mm. The ends of the beams and the slab are restrained by stubs. Although it is assumed that the torsional stiffness and drying shrinkage restraint from the stubs will be higher than in actual structures, the impact of the restraints on the structural performance of the specimens is neglected in this study. The longitudinal reinforcement ratios of the large beams, sub-beams, and slabs are 1.7%, 1.7%, and 0.4%, respectively. The longitudinal reinforcements are anchored to the stubs. The transverse reinforcements for the large beams and sub-beams are sufficient to prevent shear failure, as shown in Figure 1. The only difference between the two specimens is their curing conditions. One specimen, named the “wet specimen”, was only cured in water to avoid shrinkage, and the other specimen, named the “dried specimen”, was cured (dried) in air after curing in water.

### 2.2. Material Properties

#### 2.2.1. Concrete

The mixture proportion and properties of the fresh concrete are presented in Table 1, and the long-term concrete properties are shown in Table 2. W, C, G, S and AE in Table 1 represent water, cement, gravel, sand, and air entrainer as constituent material for concrete. *f_c_*, *f_t_*, *E_c_*, and *G_ft_* in Table 2 represent, respectively, the compressive strength, split tensile strength, Young’s modulus, and fracture energy of the concrete based on a property test. The maximum size of the aggregate was 15 mm. Compression tests and split tensile tests were conducted on cylindrical specimens with a diameter and height of 100 and 200 mm according to the Japanese Industrial Standards (JIS) (JIS A-1108, JIS A-1149, and JIS A-1113), and a fracture energy test was conducted on a 400 mm × 100 mm × 100 mm concrete specimen with a notch of 30 mm × 5 mm [38]. Property tests were performed after 110 days of curing in water for the wet specimen and after 107 days of curing in water followed by 318 days of open-air drying for the dried specimen. The open-air drying refers to standard laboratory conditions without controlling temperature and relative humidity around the specimen. The stress–strain relationships for the wet and dried concrete specimens are depicted in Figure 2a,b, respectively. The average of the five specimens for both tests is taken as the representative compressive strength of the concrete, as indicated by the dashed lines.

#### 2.2.2. Reinforcement

The rebar properties are presented in Table 3, specifying the type, purpose, and location of each piece of rebar. Four different rebar types with nominal diameters of 4, 6, 10, and 13 mm were used as longitudinal and transverse reinforcement for the slab and stirrups and as longitudinal reinforcement for the sub-beams and large beams, respectively. *E_s_*, *f_y_*, and *f_u_* are the Young’s modulus, yield strength, and ultimate strength, respectively, obtained from property tests. The values shown in Table 3 are the average of the three specimens tested under direct tensile tests. The stress–strain behavior of each piece of rebar is depicted in Figure 3. Since the yielding point is not clear except for the D10 reinforcement, the yield stress of the rebar is defined as the intersection of the stress–strain curve with a 0.2% offset in the initial stiffness according to JIS Z 2241.

### 2.3. Curing Condition

Two identical specimens, wet and dried, were cast and cured simultaneously until sufficient hydration was achieved. The wet specimen was loaded after 110 days of curing in water, as shown in Figure 4a, and the dried specimen was loaded after 107 days of curing in water and 319 days of in-air curing, as shown in Figure 4b. During the drying, the stubs were kept wet to prevent shrinkage, and restraining steel frames were installed between the two stubs, as shown in Figure 4b and Figure 5, to restrain the shrinkage as done by columns and foundations in actual structures. The restraint steel frames consisted of eight L-100 × 100 × 13 (SS400) frames located between the stubs at the four corners of the specimen, as shown in Figure 4b and Figure 5. The cross-sectional area of the steel frames was chosen so that the restraint ratio is equal to 0.6 to simulate restraint in an actual structure without considering the restraint of the rebar after shrinkage. The restraint ratio is defined as
(1)$$\lambda =\frac{\epsilon_{r}}{\epsilon_{f}}$$
where *ε_r_* is the shrinkage strain in restrained concrete, and *ε_f_* is the shrinkage strain in unrestrained concrete.

The longitudinal and transverse displacements of the specimen were measured to obtain the shrinkage strain during drying, as shown in Figure 5. The strain of the reinforcement was measured by strain gauges installed on the reinforcement, as shown in Figure 1. To understand the shrinkage behavior of unrestrained plain concrete, dummy specimens of the large beam, sub-beam, and slab (see Figure 6) were prepared with the same materials and conditions as the wet specimen. Thermocouples were installed to measure the shrinkage strain and temperature inside the concrete. The surfaces of the dummy specimen were sealed, as shown by the shaded area in Figure 6, to simulate the boundary conditions of the members. The shrinkage cracks during drying were examined once a week in the first month and once a month thereafter.

### 2.4. Loading Arrangement

The specimens were installed in the loading system shown in Figure 7. Due to the loading system, the specimens were installed vertically and fixed to the frame with high-strength steel bars for prestressed concrete through the stubs. A hydraulic horizontal jack was installed in the frame to apply the horizontal load at the center of the specimen from the out-of-plane direction (*Y*-direction in Figure 7). No axial load was applied to the specimen. The displacement in the *X*-direction and rotation about the *X*- and *Z*-axes were restrained by the loading frame placed on the specimen. The signs in Figure 7 indicate the loading direction.

Horizontal loading was applied starting from the positive direction and cyclic loading was applied repeatedly to the specimen in the following sequence: The first three cycles were controlled by loads of ±10, ±20, and ±30 kN, and cycles after the third cycle were controlled by relative displacement between the top and bottom stubs, as shown in Table 4.

## 3. Experimental Results

### 3.1. Drying and Shrinkage

During the 318 days of curing, shrinkage cracks were observed on the dried specimen as shown in Figure 8. The shrinkage cracks mainly appeared close to the stub since it is restrained. Global shrinkage was measured by the longitudinal and transverse frames, as shown in Figure 5. The global shrinkage strain, the strain in the reinforcement, and ambient temperature are compared in Figure 9, where the left and right vertical axes represent strain and temperature, respectively. Note that a negative strain represents compression of the rebar. The average shrinkage strain in the concrete in both the longitudinal and transverse directions was approximately −650 μ at the end of the 318 days, which is close to the maximum recorded strain of Rebar B (see Figure 1), as shown in Figure 9. In contrast, the strains of all other rebar were smaller and the average of the strain in the reinforcement in both the top and bottom cross-sections (see Figure 9, Figure A2 and Figure A3) near both edges was approximately 130 to 140 μ. The average shrinkage strains of the dummy specimen of the large beam, sub-beam, and slab, assumed to be representative of the shrinkage strain, were −711, −590, and −822 μ, respectively (see Figure A1), suggesting that shrinkage cracks might lead to a relaxation of the tensile shrinkage strain in the concrete, as seen in Figure 8.

### 3.2. Load–Displacement Relationship

Figure 10 shows the horizontal load–drift relationships of the wet and dried specimens. The yielding of the rebar reinforcement (as suggested by the property test results in Table 3) in the slab and large beams are denoted by circles and squares, respectively. The specimens under discussion fails in flexure through yielding of tension rebars. The cracks are mainly flexural cracks as it occurs at location of maximum flexural moment (discussed in Section 3.3) without much change in its orientation along depth with increasing load or drift. The dashed line shows the yielding and ultimate capacities as calculated by cross-sectional analyses, where the large-beam-rebar yielding capacity and maximum capacity were computed by Response-2000 using the fiber model [39]. The material parameters were defined based on the property test results shown in Table 2 and Table 3. The stress–strain relationship for concrete in compression was assumed to follow the Popovics model [40] and was assumed in tension to follow the cutoff model. In the cutoff model, the stress is linear with strain up to the tensile strength of the concrete and then the stress instantly reduces to zero. For the reinforcement, a bilinear model was assumed in both tension and compression. A moment can be positive at the top (slab in tension) and negative at the bottom (slab in compression) for positive loading and vice-versa for negative loading, as shown in Figure 7, and due to the asymmetrical cross section of the specimen, the moment capacity differs between those two sections for a given loading. The horizontal load-carrying capacity *V* is estimated from the positive and negative moment capacities *M*^+^ and *M*^−^ as
(2)$$V=\frac{M^{+}+M^{-}}{l}$$
where, l is the clear span of the specimen.

The maximum load-carrying capacities of the wet and dried specimens from the experimental results were 250 and 275 kN, respectively, which is close to the ultimate capacity of 262 kN obtained from the sectional analysis. This indicates that the drying does not affect the ultimate capacity.

Figure 11 compares the skeleton curves of the specimens with first and full (all rebar) yielding of the reinforcement of the large beams (LB), sub-beams (SB), and slabs. The two dashed lines represent the estimated tri-linear load–drift relationship of the beam–slab members (Figure 11) based on Architectural Institute of Japan (AIJ) and American Concrete Institute (ACI) standards as shown in Figure 12. The initial lateral structural stiffness *K*_0_, crack capacity *V_cr_*, yield capacity *V_y_*, and effective yield stiffness *K_y_* are given as follows.

The initial lateral structural stiffness *K*_0_ can be estimated based on the elastic bending theory of a beam as
(3)$$K_{0}=\frac{12E_{c}I_{e}}{{l^{\prime }}^{3}}$$
where, $E_{c}$ is the Young’s modulus of concrete, $I_{e}$ is the moment of inertia of the cross-section, including that of rebar, and l’ is the effective span. In the AIJ and ACI standards, the different effective lengths $l^{\prime }$ are specified as
(4)$${l^{\prime }}_{AIJ}=l+\frac{D}{2}$$
(5)$${l^{\prime }}_{ACI}=l+D$$
respectively, where l is the clear span and *D* is the entire beam depth. The additional lengths *D*/2 and *D* are different between the standards and correspond to the reduction of the elastic stiffness of members at both edges due to the degradation of the rotational stiffness at the cross section near both edges.

The estimated lateral structural stiffness is compared to the experimental lateral structural stiffness in Table 5. As shown in Table 5 and Figure 11, the experimental lateral structural stiffness of the wet specimen nearly corresponds to the estimation from the ACI standards while the estimation from the AIJ standards overestimates the test results. In the experimental results, the lateral structural stiffness of the specimen with drying decreased to 77% of that of the wet specimen in the 30 kN cycle. In a past study on an RC shear wall, the stiffness reduced to 54% after drying [34]. This differing reduction depends on the type of RC member and the governing mechanism, as discussed in Section 4.

The flexural crack capacity *V_cr_* was calculated from the average positive and negative crack moments (*M_cr_*^+^ and *M_cr_*^−^) from both standards as
(6)$$V_{cr}=\frac{M_{cr}^{+}+M_{cr}^{-}}{l}$$
(7)$$M_{cr}=Z_{e}f_{t}$$
(8)$$Z_{e}=\frac{I_{e}}{h_{t}}$$
and $h_{t}$ is the distance from the extreme tension fiber to the neutral axis.

Similarly, the yield capacity *V_y_* was calculated from the average positive and negative yield moments (*M_y_*^+^ and *M_y_*^−^) in both standards as
(9)$$V_{y}=\frac{M_{y}^{+}+M_{y}^{-}}{l}$$
(10)$$M_{y}=\overset{n}{\underset{i=1}{\sum }}{\left(A_{s}f_{y}j\right)}_{i}$$

*A_s_* is the area of longitudinal tension reinforcement of each component (assumed as zero for the slab when the slab is in compression), *j* is the lever arm length (0.9*d* for the large beam and 0.9*d* − *d_s_* for the sub-beam and slab), *d* is the effective depth, and *d_s_* is the distance between the slab and sub-beam’s tension rebar, from the tension rebar in the large beam, *n* is total number of components.

For the effective yield stiffness, *K_y_* is specified in the AIJ standard as the product of the initial lateral structural stiffness and the stiffness reduction factor *α_y_* [41]:(11)$$K_{y}=\alpha_{y}K_{0}$$
(12)$$\alpha_{y}=\left(0.043+1.64np_{t}+0.043l/2D+0.33\eta_{0}\right){\left(\frac{d}{D}\right)}^{2}$$
where *n* is the modular ratio, *p_t_* is the ratio of *A_s_* to *bd*, *b* is the width of each component, and $\eta_{0}$ is the axial load ratio (zero for beams). For the specimen in this study, the stiffness reduction factor is 0.18, whereas 0.35 is recommended for beams in the ACI standard.

Therefore, the drift at the yielding point Δ*_y_* is given by *V_y_*/*α_y_K*_0_*ℓ*. Although the contribution of the effective width to the lateral structural stiffness and capacity is considered in the AIJ standards, here, the contribution of the whole slab is considered, owing to the presence of the stubs. As shown in Figure 11, the estimated lateral structural stiffness is equal to that as calculated by the cross-sectional analyses.

The yield drift nearly equal for both specimens, even though a lower load-carrying capacity was observed in the dried specimen before yielding. For both specimens, the reinforcement of the slab yields first, followed by yielding of the rebar of the sub-beam and large beams. However, yielding of rebars of dried specimen particularly slab rebars is delayed. The first yielding of the slab rebar is observed at a drift of 0.25% for both specimens; the load-carrying capacity for the wet specimen is approximately 125 kN, whereas that for the dried specimen is 105 kN, as shown in Figure 10. However, this difference in load-carrying capacity gradually reduces until the final yielding point of the specimen, as can be seen in Figure 10 and Figure 11. All the rebar yielding points occurred at around 1% to 1.33% drift for both specimens. Both the yielding capacity and ultimate capacity showed good agreement with the calculations from the AIJ and ACI standards. The drift at which the reinforcement of the large beam yields corresponds to the estimation from the AIJ standard.

### 3.3. Crack Patterns

The first flexural crack was observed at 30 kN for both specimens in the loading test, and after the continuous loading cycles, more cracks appeared and propagated. The crack pattern observed after the cycle with 2% drift is shown in Figure 13. The dashed, blue, and red lines represent cracks from shrinkage, positive loading, and negative loading, respectively. The cracks occurring on the specimen due to drying and loading are sketched on the paper and then transferred to AutoCAD (Student version, 2019, Autodesk, California City, CA, USA). Note that the positive-loading flexural moment results in tension and compression at the top and bottom of the slab, respectively, and vice versa for negative loading, as shown in Figure 7. Hence, the flexural cracks are concentrated to the top and bottom of the specimen and are mostly parallel to the critical section (specimen–stub interface) as it can be seen in Figure 13. Some differences between the specimens in the crack pattern due to drying were observed, namely that several transverse cracks appeared gradually with increasing drift from both fixed ends to the center of the slabs through the sub-beam in the dried specimen. In contrast, for the wet specimen, diagonal cracking was first observed at the 0.5% drift cycle. This might be explained by the twisting behavior of the large beam and slab due to the absence of shrinkage cracks.

## 4. Discussion

As shown in the previous section, the impact of drying is indicated by the lateral structural stiffness, crack distribution, and reinforcement yielding. In this section, the mechanism for this impact is discussed in terms of the flexural stiffness as affected by shrinkage cracks.

As a result of the drying, in the load–drift relationship, a difference in the lateral structural stiffness was observed, as shown in Table 5. Therefore, the lateral structural stiffness estimated from the specifications does not correspond to the test results from the dried specimen. Furthermore, as seen in the crack distribution, several transverse cracks appeared on the slab in tension only after load application, i.e., for the slab in compression, the cracks do not appear or are closed in both specimens, regardless of drying. Thus, the shrinkage cracks affect the lateral structural stiffness for the slab in tension only and not the stiffness for the slab in compression, which may lead to an overestimation of the lateral structural stiffness of the dried specimen.

To verify the differences in lateral structural stiffness for the slab in compression and tension, the experimental flexural stiffness is discussed. Since this can be obtained from the *M*–*ϕ* relationship, the flexural moment at a cross section at the edges of the slab is estimated to be the product of horizontal load and contraflexure height. Assuming the strain distribution is linear between 450 mm and the mid-height of the specimen, the contraflexure point is identified as the point of zero strain along the height of the specimen from the stub. Figure 14 shows the calculated contraflexure height. There is some variation in the smaller loading cycle; however, the calculated contraflexure height converges to the center of the specimen height with increasing drift. With this contra-flexural height and horizontal load, the flexural moment at the top and bottom edges of the slabs was calculated for each cycle.

As shown in Figure 15, the curvature at the cross section at the edges of the specimen can be obtained using the measured strains of the reinforcement of the large beam and the distance between reinforcement as
(13)$$\phi =\left(\varepsilon_{1}-\varepsilon_{2}\right)/d_{1}$$
where *ε*_1_ is the strain of the top reinforcement, *ε*_2_ is the strain of the bottom reinforcement, and *d*_1_ is the distance between the top and bottom reinforcement.

The *M*−*ϕ* relationship for the wet and dried specimens is compared with the sectional analysis results for the slab in compression and tension in Figure 16a,b, respectively. The flexural stiffness at 10 and 30 kN is summarized in Table 5 It is found that the sectional analysis results correspond to the test results for the wet specimen. For the slab in tension, the flexural stiffness of the dried specimen under 10 and 30 kN is approximately 50% of that of the wet specimen, while in for the slab in compression, the flexural stiffness of the dried specimen is approximately 60% of that of the wet specimen under 10 kN. At 30 kN, for the part of the specimen where the slab is in compression, the shrinkage cracks are closed resulting in slab full section being effective to the flexural stiffness. Therefore, the moments of inertia in compression for the wet and dried specimens are the same, so there is no change in flexural stiffness. From this evidence, it is concluded that any cracks (flexural and shrinkage) do not significantly affect the flexural stiffness for a slab in compression, which may lead to a difference in the decrease in lateral structural stiffness (Table 5) and flexural stiffness (Table 5), i.e., the shaded cross-sectional area of the slab shown in Figure 17 does not contribute to the moment of inertia for a slab in tension. Following this, a sectional analysis neglecting concrete in the slab was performed (dashed line in Figure 16a), and the decrease in the average flexural stiffness for the slab in compression and tension is closer to the decrease in the experimental lateral structural stiffness, as shown in Table 5.

The decrease in the lateral structural stiffness ratio between the two specimens to 0.77 as shown in Table 5 is higher than the decrease in the average flexural stiffness ratio to 0.64 as shown in Table 5. The drying cracks that appeared near the cross section at the edge affected the curvature distribution of the specimen near the edges only. If the drying cracks occurred evenly over the specimen with significantly further drying, then the decrease in the lateral structural stiffness will be close to the decrease in flexural stiffness.

Although a considerable difference in the initial lateral structural stiffness was observed between the wet specimen and dried specimen, the changes in final yield stiffness and final yield capacity were not clear. For the slab in tension for the dried specimen, the drift exhibited at yielding was similar to that of the wet specimen. In contrast, for the slab in compression, the yielding points for the wet and dried specimens in the experiment and analysis were nearly identical. As Achiha et al. (2020) [42] concluded that if drying shrinkage strain is less than cracking strain, then residual stress in the rebar leads to early yielding of the rebar in compression and delayed yielding of tension rebar, it is assumed that the shrinkage strain in the rebar is exhibited during curing. However, when the shrinkage strain of the concrete reaches its tensile strain, the stress is released by the occurrence of cracks and this may result in a lack of influence on the yielding of the specimen.

## 5. Conclusions

In this study, to investigate the effect of changes in material properties due to drying on the mechanical characteristics of reinforced concrete (RC) beam–slab members, cyclic loading tests were conducted on two RC beam–slab specimens cured under two different conditions: with and without drying. From a comparison of the experimental results for the two specimens, the following conclusions are obtained:Although a maximum drying shrinkage strain of −650 μ developed in the specimen cured in the open air, the strain became approximately one-fifth of the average value due to shrinkage cracks that appeared during curing.The lateral structural stiffness of the specimen with drying was decreased to 77% of that of the specimen without drying. The reduction of the lateral structural stiffness due to shrinkage was verified through the calculation of the flexural stiffness, considering that the drying shrinkage decreases the moment of inertia of the slab in tension but not in compression (since the shrinkage cracks are thin, cracks get closed even in small compression).The reduction ratio of the lateral structural stiffness of the dried specimen to that of the wet specimen does not linearly correlate to the ratio of the flexural stiffness, due to the location of shrinkage-induced cracks that were formed around the edges of the specimen where the specimen was connected to the stub.For both specimens and with and without drying, no difference was observed in the drift where the reinforcing bars in the main beams yielded. This is because the drying shrinkage-induced cracks released the accumulated tensile stress in the concrete, and the self-balancing-stress in the reinforcing bars was released or easily released during the cyclic loading process.

Mechanism of impact of drying as concluded in this study is well applicable to similar kind of structures. Therefore, drying can be considered for reinforced concrete structures design through this mechanism for a serviceable and durable structure.

## Figures and Tables

**Figure 1 materials-14-01887-f001:**
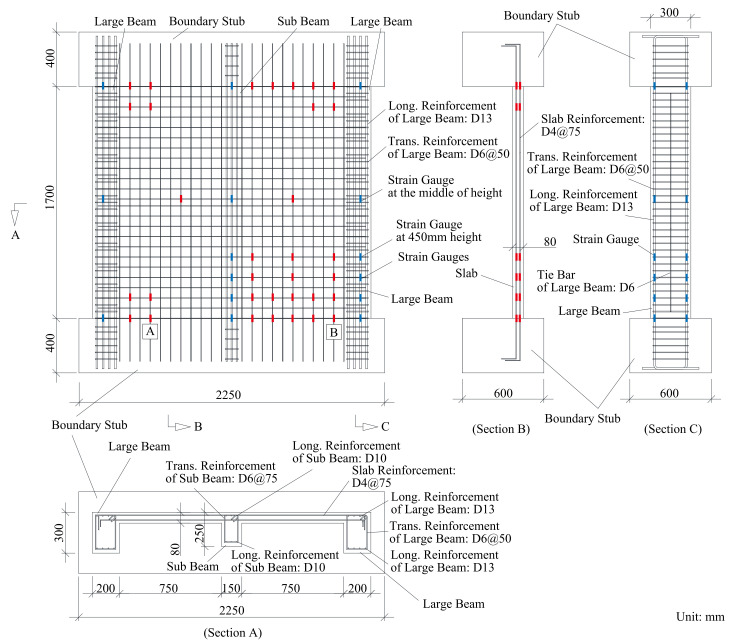
Schematic of the specimens.

**Figure 2 materials-14-01887-f002:**
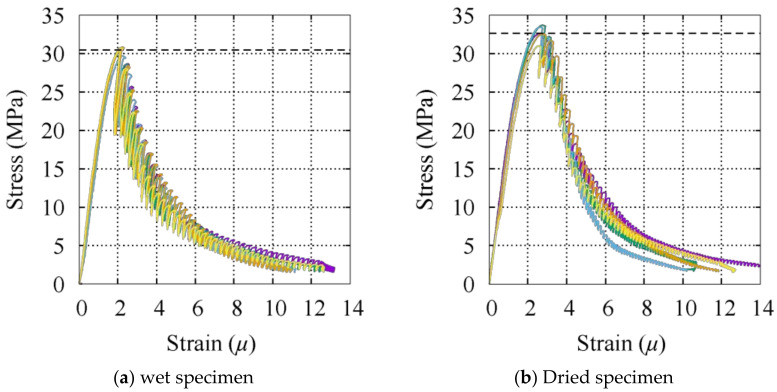
Stress–strain relationships for the concrete in compression. (**a**) Wet specimen, (**b**) dried specimen.

**Figure 3 materials-14-01887-f003:**
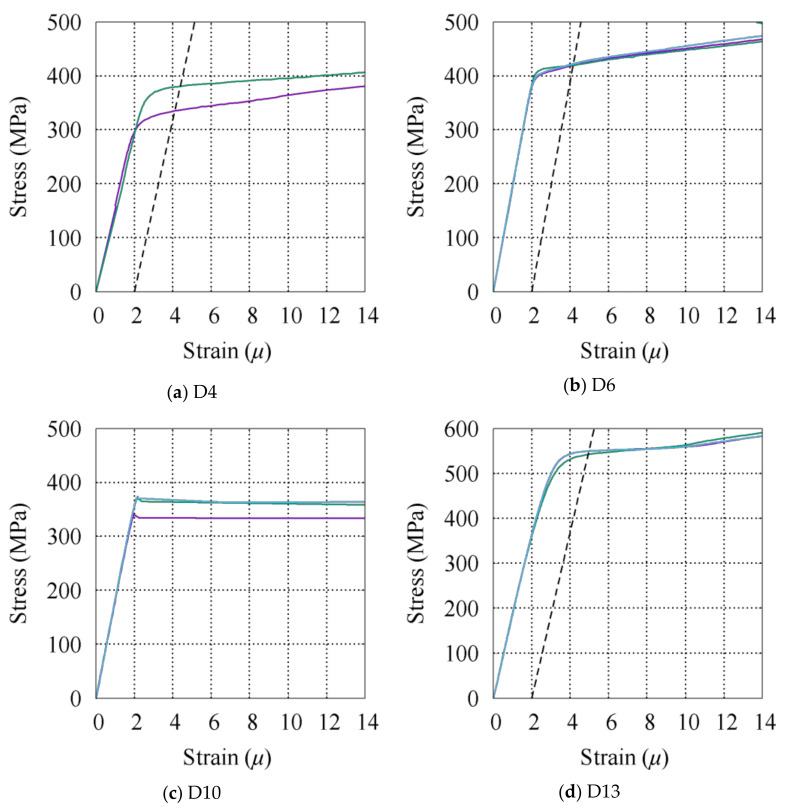
Rebar tension test results. (**a**) D4, (**b**) D6, (**c**) D10, (**d**) D13.

**Figure 4 materials-14-01887-f004:**
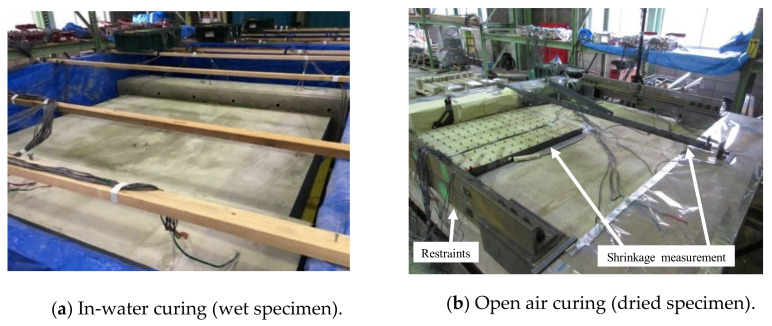
Curing conditions. (**a**) In-water curing (wet specimen), (**b**) open air curing (dried specimen).

**Figure 5 materials-14-01887-f005:**
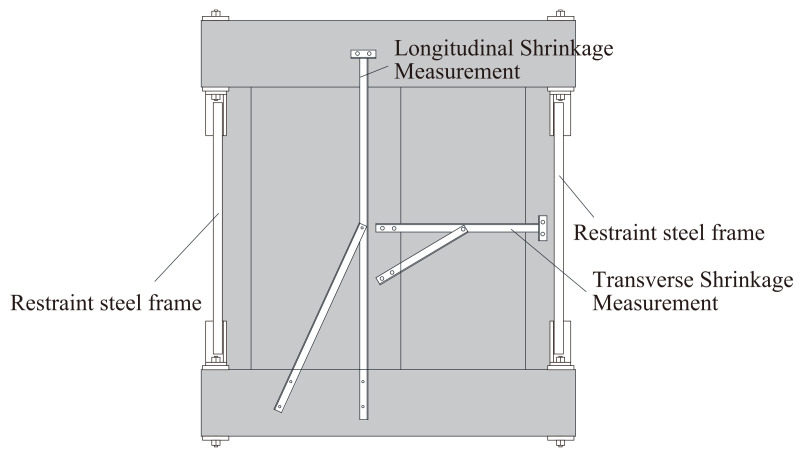
The measurement of the shrinkage displacement.

**Figure 6 materials-14-01887-f006:**
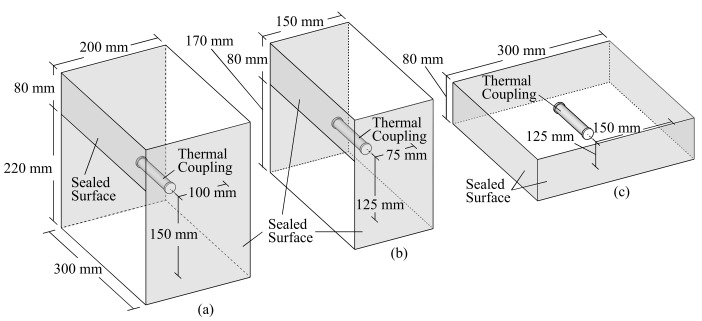
The dummy specimens: (**a**) large beam, (**b**) sub-beam, and (**c**) slab.

**Figure 7 materials-14-01887-f007:**
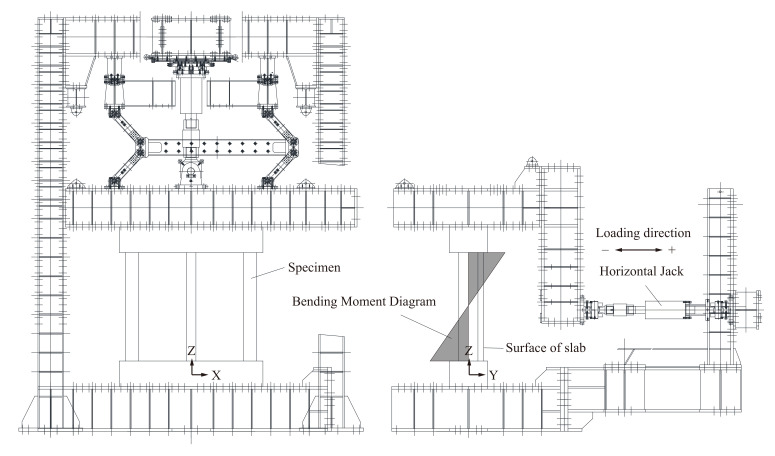
Schematic of the loading arrangement.

**Figure 8 materials-14-01887-f008:**
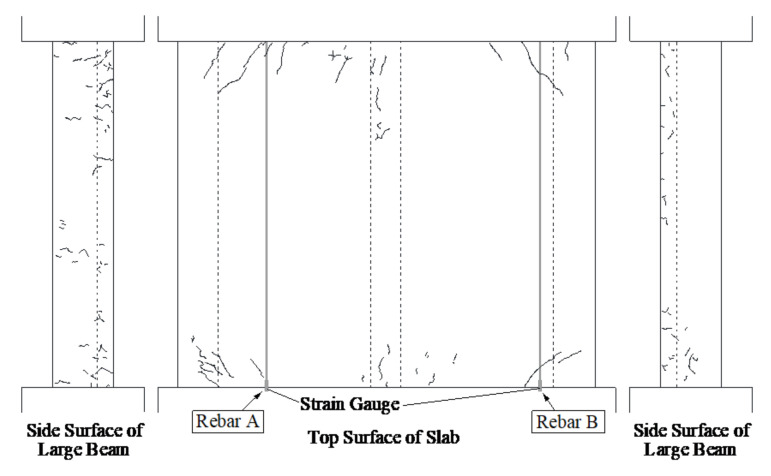
Shrinkage cracks after 318 days of drying (A and B show the locations of rebars that correspond to those in Figure 1 and Figure 9).

**Figure 9 materials-14-01887-f009:**
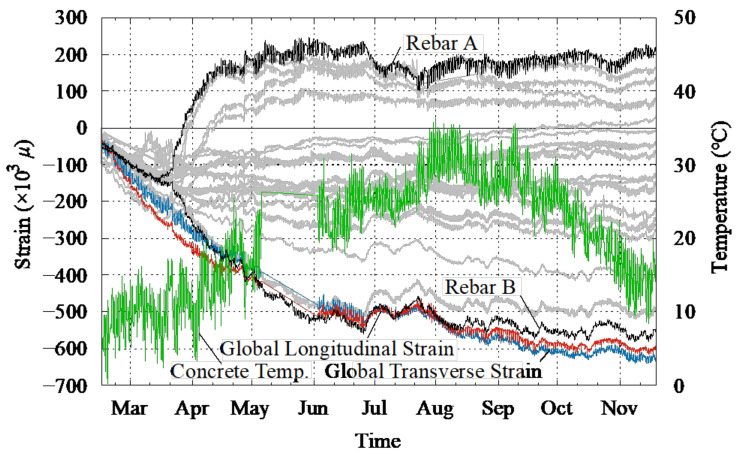
Shrinkage strain, rebar strain at the bottom interface, and temperature history during drying (* Location of Rebar A and B are shown in Figure 1 and Figure 8.).

**Figure 10 materials-14-01887-f010:**
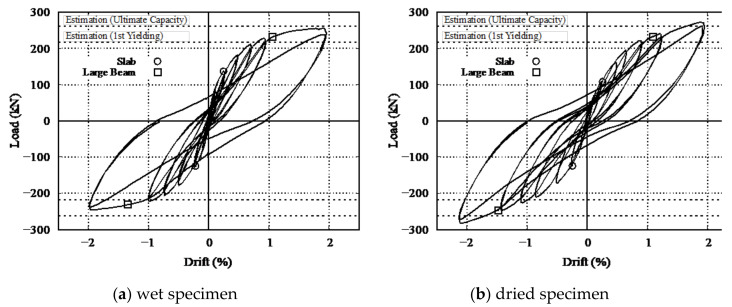
Load–drift relationships. (**a**) Wet specimen, (**b**) dried specimen.

**Figure 11 materials-14-01887-f011:**
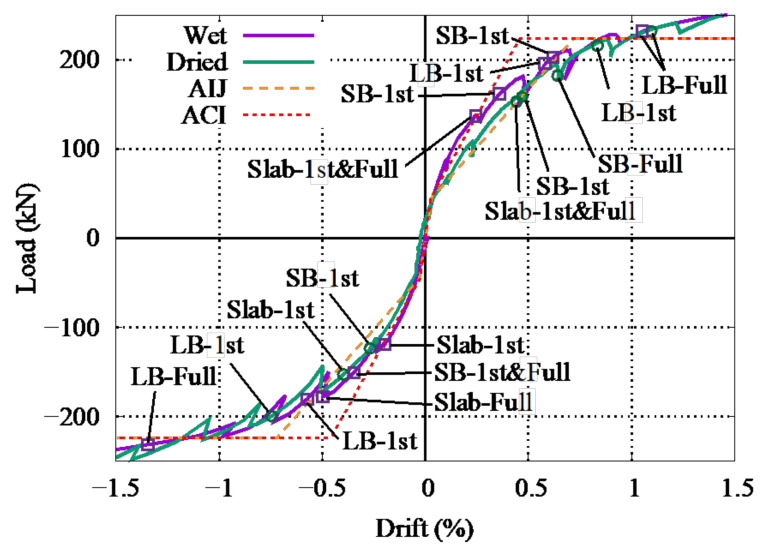
Comparison of the skeleton curves for the experimental and estimated results.

**Figure 12 materials-14-01887-f012:**
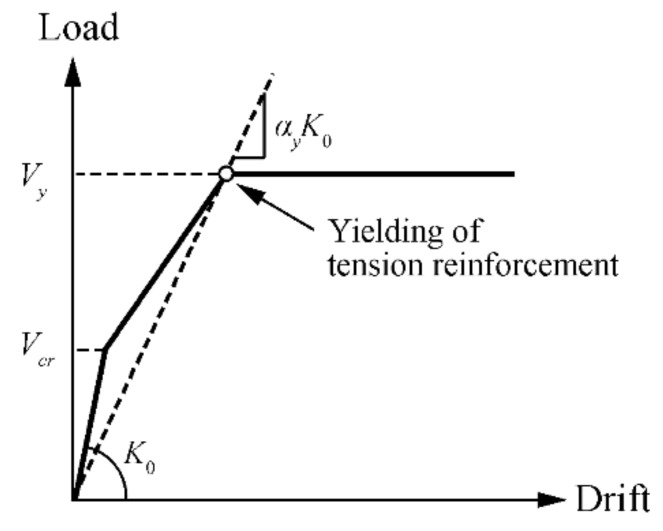
Tri-linear load–drift relationship.

**Figure 13 materials-14-01887-f013:**
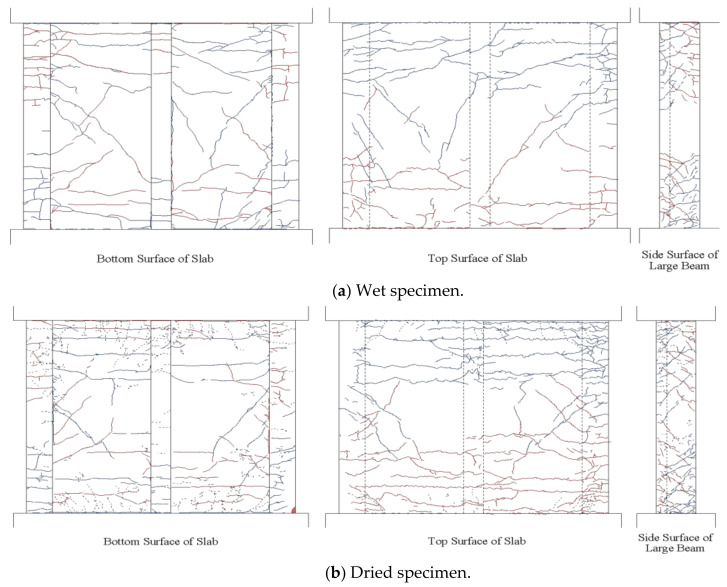
Crack patterns. (**a**) Wet specimen, (**b**) dried specimen.

**Figure 14 materials-14-01887-f014:**
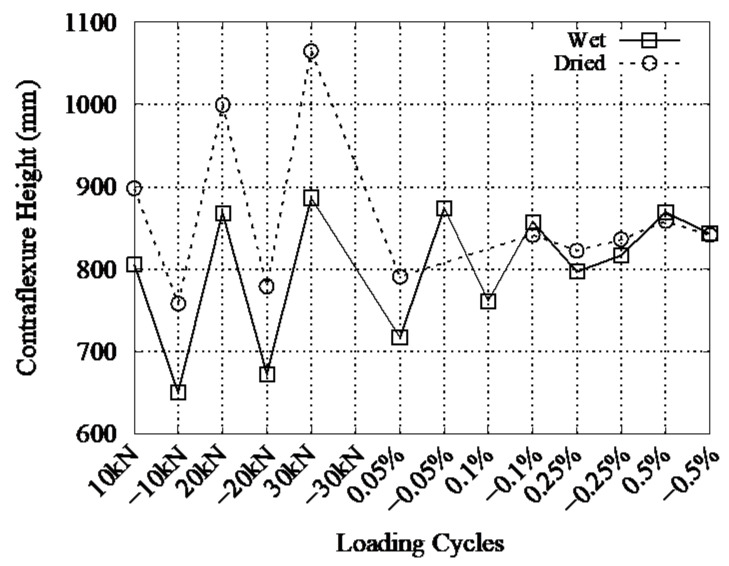
Contraflexural height at each load peak.

**Figure 15 materials-14-01887-f015:**
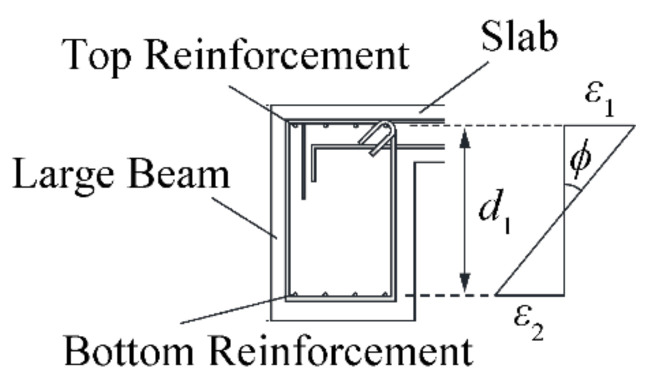
Curvature considering strain in the rebar.

**Figure 16 materials-14-01887-f016:**
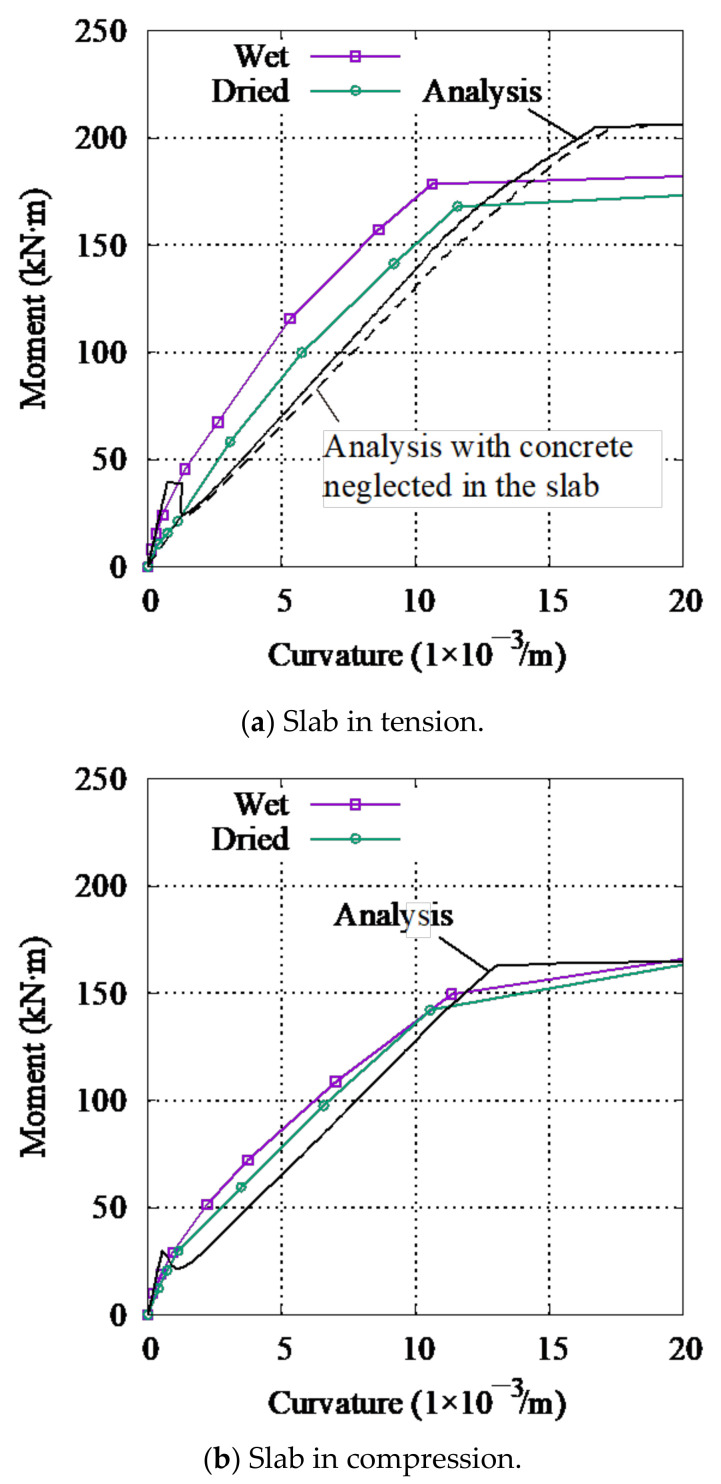
*M*−*ϕ* relationship. (**a**) Slab in tension, (**b**) slab in compression.

**Figure 17 materials-14-01887-f017:**
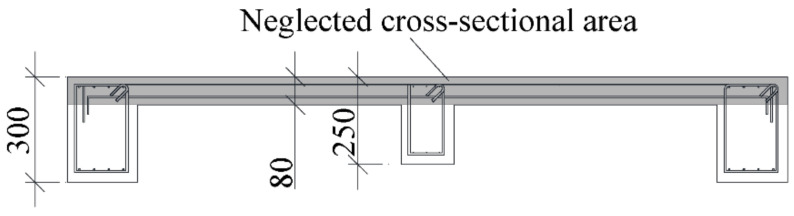
The cross section after drying considered in the calculation.

**Table 1 materials-14-01887-t001:** Mixture proportion and properties of the fresh concrete.

Mixture Proportion							Fresh Properties		
W/C (%)	s/a (%)	Mass (kg/m^3^)					Slump (mm)	Air (%)	Temp. (°C)
		W	C	G	S	AE *			
60.8	51.2	182	300	854	885	3	180	4.5	8

* AE is Air Entrainer.

**Table 2 materials-14-01887-t002:** Concrete properties.

Specimen	Curing Period	*f_c_* (MPa)	*f_t_* (MPa)	*E_c_* (GPa)	*G_ft_* (Nm)
Wet Specimen	Concrete specimens for property tests: tested after 110 days of curing in water	30.5	2.85	25.7	107.9
	Beam–slab specimen: loaded after 90 days of curing in water plus 10 days of preparation				
Dried Specimen	Concrete specimens for property tests: tested after 107 days of curing in water plus 318 days of drying	32.6	2.47	23.7	136.1
	Beam–slab specimen: loaded after 107 days of curing in water plus 319 days of drying and preparation				

**Table 3 materials-14-01887-t003:** Rebar properties.

Rebar Type	Location	Type of Steel	*E_s_* (GPa)	*f_y_* (MPa)	*f_u_* (MPa)
D4	Slab	SD345	170	357	489
D6	Stirrups	SD345	161	420	682
D10	Sub-beam	SD345	173	355	514
D13	Large beam	SD490	178	545	734

**Table 4 materials-14-01887-t004:** Load application cycles.

Loading Control		Load Controlled			Drift Controlled								
Load (kN)/Drift (%)		10	20	30	0.05	0.1	0.25	0.5	0.75	1	1.3	2	4
Number of Cycles	Wet specimen	1	1	1	1	1	2	2	0	2	0	2	2
	Dried specimen	1	1	1	0	1	1	1	1	2	2	2	2

**Table 5 materials-14-01887-t005:** Initial lateral structural stiffness K (kN/mm).

Load	Experimental Lateral Stiffness			Estimated Lateral Stiffness, *K_0_*	
	Wet	Dry	Dry/Wet	AIJ	ACI
10 kN	78	59	0.76	104	82
30 kN	70	54	0.77		

## Data Availability

Data sharing is not applicable to this article.

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
