# Peer review of "Impact of Drying on Structural Performance of Reinforced Concrete Beam with Slab"

_materials, 2021, doi:10.3390/ma14081887_

Round 1
Reviewer 1 Report
The authors examined the impact of dying on structural performance of reinforced concrete beam with slab. The topic is interesting, and the-state-of-art provided is well described. The research methodology is appropriate and well described. The experimental results are also well presented and discussed. The conclusions are well justified with the results. English use and formatting are good. Moreover, the bibliography is related to the topic and extent enough.
Author Response
Thank you for your review. I attach our responses.

Reviewer 2 Report
The authors prepared an interesting study dealing with a practical topic. It is clear that they are very familiar with the issue solved. The experimental program is designed and carried out properly. Results are clear and well discussed. The text is well written and easy to follow. I have only a few remarks which can help to improve the study, these are itemized below:
- Introduction: reduce the number of references listed in one link (see for example the first paragraph on the second page: [23-30]). Cited studies should be selected into relevant groups and discussed. Moreover, their results should be mentioned briefly. It is not possible to only describe that some authors dealt with similar issues without indicating their findings (such a problem is noticeable especially in the second paragraph). The whole introduction is done a little carelessly when compared to the other chapters.
- What is “AE” in Table 1?
- Consider whether to specify what is open-air drying (page 3 and further). I recommend to describe that it is talked about standard laboratory conditions.
- Revise properly all references to figures through the text. I often met bad links.
- Paragraph 3.3: how was the crack pattern observed (depicted in Figure 12 and also in 7)?
Author Response
Thank you for your time and review. We attach our responses.

Reviewer 3 Report
The manuscript deals with experimental testing to assess shrinkage influence in the structural response of a RC beam+slab system. Given is overall quality and significance it is rocemmended to be publishes after minor improvements.
Some guidance in such improvements is provided below.
The findings part of the abstract is confusing and may be improved.
The experimental program is of merit and the research interest is duly justified in the introduction.
Table 1: Information about the meaning of W, C, S, G and AE is missing.
Table 2: The concept of “days after casting and Information about it are confusing and shall be clarified
Figure 1: It is suggested to make clear that the depicted behavior is under compression.
Figure 2: It seems that the criterion was to limit the plotted curves to 14 microstrains. Therefore, the ultimate strength is not depicted.
Ln 157 and Figure 4: consider changing “vertical” and “horizontal” to longitudinal and transverse if considered appropriate.
Except for Figure 1, figure numbering is wrong.
Ln 188 The existence of a vertical jack in the figure generates some confusion as it is stated that “no axial load is applied”.
Table 5: it is interesting to notice that the stiffness ratio between AIJ-estimated and wet is similar to that of dry to wet.
Author Response

(The authors gave the same response as above.)

Reviewer 4 Report
Comments
This paper studied the effect of drying on structural performance of reinforced concrete beam with slab. The outcome is interesting for readers. However, there are several aspects that need to be improved. The reviewer can only recommend for publication if the author satisfactorily address the following comments in the revised version.
- The author mentioned “drying shrinkage decreases the moment of inertia of the slab in tension but not in compression”. On what basis this assumption was made.
- Figure 12. How the author obtained this crack patterns? Is it captured by the image analysis of the original crack patterns or it was sketched? Need a clear statement in the manuscript.
- 10, the lower range of limit is missing.
- How the stiffness reduction factor equation (Eq. 12) was obtained?
- How many specimens were tested for each case?
- The failure mechanism of the specimen should be discussed more clearly.
- The novelty of the study should be highlighted at the end of introduction section. How this study is different from the published study in literature?
- How the outcome of this study will benefit researchers and end users? This need to be highlighted in introduction or end of conclusion.
- The background study on the effect of drying shrinkage on concrete need to be improved. Recently, it was found that drying shrinkage affect the durability [Ref: 3D-printed concrete: applications, performance, and challenges]and microstructural properties [Ref: Characteristics, strength development and microstructure of cement mortar containing oil-contaminated sand]. Suggest to include them in introduction section with proper citations to improve the background study.
I would be happy to see the revised version to understand how these comments are being addressed.
Author Response

(The authors gave the same response as above.)

Round 2
Reviewer 4 Report
I have no further comments